# Assessing Functional Independence and Associated Factors in Older Populations of Kazakhstan: Implications for Long-Term Care

**DOI:** 10.3390/healthcare13151878

**Published:** 2025-07-31

**Authors:** Gulzhainar Yeskazina, Ainur Yeshmanova, Gulnara Temirova, Elmira Myrzakhmet, Maya Alibekova, Aigul Tazhiyeva, Shynar Ryspekova, Akmaral Abdykulova, Ainur Nuftieva, Tamara Abdirova, Zhanar Mombiyeva, Indira Omarova

**Affiliations:** 1Faculty of Medicine and Health Care, Department of Health Policy and Public Health, Kazakh National University Named After Al-Farabi, Al-Farabi 71, Almaty 050069, Kazakhstan; 2Department of Nursing, Aktobe Higher Medical College Named After Hero of the Soviet Union Manshuk Mametova, Aktobe 030006, Kazakhstan; 3Department of Simulation Center, Marat Ospanov West Kazakhstan Medical University, Aktobe 030006, Kazakhstan; 4Department of General Medical Practice, Kazakh National Medical University Named After S.D. Asfendiyarov, 94 Tole bi, Almaty 050069, Kazakhstan; 5Department of Histology, Marat Ospanov West Kazakhstan Medical University, 68 Maresieva Street, Aktobe 030006, Kazakhstan; 6Department of Nursing, Kazakh National University Named After Al-Farabi, Almaty, Al-Farabi 71, Almaty 050069, Kazakhstan; 7Department of Oncology, Marat Ospanov West Kazakhstan Medical University, 68 Maresieva Street, Aktobe 030006, Kazakhstan; 8Department of Normal Physiology, Kazakh National Medical University Named After S.D. Asfendiyarov, 94 Tole bi, Almaty 050069, Kazakhstan; 9Department of Clinical Disciplines, International Medical School, University of International Business Named After K. Sagadiev, Almaty 050069, Kazakhstan; 10Department of Physical Medicine and Rehabilitation, Sports Medicine, Kazakh National Medical University Named After S.D. Asfendiyarov, 94 Tole bi, Almaty 050069, Kazakhstan

**Keywords:** Barthel Index, long-term care, functional dependence, risk factors

## Abstract

**Background/Objectives**: Accurately assessing the independence level of older adults using useful assessment tools is an important step toward providing them with the necessary care while preserving their dignity. These tools allow older adults to receive effective, personalized home care, which improves their quality of life. This study aimed to clarify the current prevalence of severe and complete functional dependence and associated factors among Kazakhstan’s older adults aged >60 years. **Methods**: This cross-sectional study was conducted in several polyclinics and geriatric service care centers in two cities of Kazakhstan from March to May 2024. Functional status was assessed by the Barthel Index. We combined the selection into two categories: total dependency and severe dependency in the category “dependent”, and moderate dependency, slight dependency, and total independence in the category “active patients”. **Results**: Among the 642 older people in this study, 43.3% were dependent patients, and 56.7% were active patients. The odds of severe and total functional dependence are significantly higher for frail participants (adjusted odds ratio (AOR) = 2.96, 95% confidence interval (CI) [1.70, 5.16], *p* < 0.001) compared to those that are not frail; eleven times higher for those at home (AOR =11.90, 95% CI [5.77, 24.55], *p* < 0.001) than those in nursing homes; two times higher for participants with sarcopenia (AOR =2.61, 95% CI [1.49, 4.55], *p* < 0.001) compared to those with no sarcopenia; and three times higher for participants with high risk of fracture (AOR =3.30, 95% CI [1.94, 5.61], *p* < 0.001) compared to those with low risk. The odds of having severe and total functional dependence are significantly higher for participants with low dynamometry (AOR =1.05, 95% CI [1.03, 1.07], *p* < 0.001) compared to those with normal dynamometry. **Conclusions**: Old age, low dynamometry (for men ≤ 29 kg, for women ≤ 17 kg), frailty, being at home, high risk of fracture and osteoporosis, and sarcopenia were associated with increased risk of severe and total functional dependence.

## 1. Introduction

As the global population ages, the need for long-term care (LTC) services, especially home care, has increased dramatically. Home care is widely recognized as a vital component of elder care, providing patients with the support they need to live independently in the comfort of their own homes. This model has gained significant traction globally due to its ability to promote patient autonomy, improve quality of life, and reduce healthcare costs. At the same time, the presence of aging populations across the world has created new challenges for healthcare systems, particularly in regions with rapidly expanding elderly demographics, such as the Commonwealth of Independent States (CIS) and Kazakhstan [1].

Globally, the aging population is expanding rapidly. According to the United Nations, the number of people aged 60 years or older is expected to increase from 1 billion in 2020 to 2.1 billion by 2050, representing nearly 22% of the global population. As a result, the need for LTC is expected to rise, with around 15% of older adults requiring some form of LTC by 2030 [2]. LTC encompasses a range of services aimed at providing ongoing support to individuals who experience a decline in function due to chronic illness or physical or mental disabilities. The nature and extent of LTC services vary based on the specific needs of the individuals, involving both direct hands-on care and general oversight. LTC aids older adults in two primary areas: activities of daily living (ADLs) and instrumental activities of daily living (IADLs). ADLs involve essential tasks such as eating, bathing, dressing, transferring between bed or chair, and using the toilet. IADLs refer to more complex activities that help sustain independence, including meal preparation, medication management, grocery shopping, and utilizing transportation [3].

Decline in the ability to perform ADLs, or “functional decline”, is a major health concern among aging populations, partly because loss of physical function is one of several reasons older adults receive institutionalized care [4].

Aging diminishes the physical and cognitive abilities of older adults, which further impacts their ADLs. Continuous clinical monitoring of functional independence in ADLs has become essential in community-based elderly care. Several reliable assessment tools have been developed to track functional changes over time. The Barthel Index (BI) is one of the most widely used scales for evaluating functional independence, particularly in assessing progress during rehabilitation. The BI scale evaluates individuals’ ability to perform 10 activities (e.g., feeding, bathing, dressing, etc.), with a total score ranging from 0 to 100. A lower BI score indicates a higher likelihood of future disability, longer recovery times, and greater care requirements [5]. Several international studies have demonstrated the effectiveness of the BI in identifying older adults at risk of requiring LTC. For example, research in Japan and China has shown that BI scores can predict LTC insurance eligibility and functional dependency [6,7], while studies in Ireland and Hong Kong have linked BI scores to care transitions and health outcomes [8,9]. These findings support the use of the BI as a simple, validated tool for early screening and stratification of functional decline. However, to date, no published data are available on the use of the BI for this purpose in Kazakhstan, a country currently undergoing rapid demographic aging. Given Kazakhstan’s evolving health and social care infrastructure, applying an internationally recognized tool such as the BI could provide evidence-based guidance for planning elder care services. Therefore, evaluating its utility in the Kazakhstan context is essential to inform national LTC strategies, identify at-risk populations earlier, and support cost-effective resource allocation.

In the CIS, the demand for LTC is on the rise. The CIS includes countries such as Russia, Kazakhstan, and Belarus, all of which are experiencing rapid demographic shifts. According to a report by the United Nations Economic Commission for Europe, the elderly population in CIS countries, defined as those aged 60 and over, was estimated at around 20% in 2020, with projections indicating further increases in the coming decades [10].

In Kazakhstan, the elderly population is growing, with people aged 60 and over accounting for approximately 14% of the total population in 2022 [11]. As the population ages, the demand for home care services has increased, especially among individuals with chronic diseases and disabilities. In 2021, it was estimated that 12% of Kazakhstan’s elderly population required LTC, and a large proportion of these individuals preferred receiving care at home [12]. In the past few decades, the proportion of the population aged 65 and above has doubled on average across Organisation for Economic Co-operation and Development countries, rising from under 9% in 1960 to 18% in 2021 (242 million people aged 65 and above, including more than 64 million who were at least 80 years old) [13]. Kazakhstan is transitioning into a demographic profile typical of aging societies but still enjoys a demographic advantage compared to the most aged nations [1]. As the proportion of elderly people increases, healthcare demands and LTC needs will become more pressing. It is time to plan and implement healthy aging strategies, leveraging experiences from advanced economies.

Despite this rising demand, LTC services in Kazakhstan are still underdeveloped. While the government has introduced reforms to address this need, challenges related to the coordination of services, availability of skilled caregivers, and public awareness remain barriers to the effective implementation of LTC models. The rapidly aging global population presents a significant challenge to healthcare systems, especially in relation to the growing demand for LTC. This demand is expected to rise sharply over the next few decades, with the elderly population increasing substantially across many regions, including in countries of the CIS and Central Asia. The development of LTC models, particularly for elderly individuals with complex medical and social needs, is becoming a critical area of focus. LTC, which includes a combination of healthcare and social support, is a key solution for improving the quality of life of older adults, enabling them to age at home with dignity and autonomy [2].

Globally, the shift toward LTC is driven by a combination of demographic trends, patient preferences, and the increasing recognition of the effectiveness of models. In both the CIS and Kazakhstan, expanding LTC services is crucial to addressing the needs of aging populations and ensuring that elderly individuals receive the care they need while maintaining their dignity and independence. The accurate assessment of dependency using tools like the BI is an essential step in delivering personalized, effective home care that can improve the quality of life of elderly individuals.

Accurate dependency assessments using useful assessment tools are essential for providing effective, personalized home care that improves the quality of life of older adults. In this context, developing, implementing, and evaluating innovative LTC models in geriatric practice is paramount. This study aimed to determine the prevalence of severe and complete functional dependence and its associated factors among Kazakhstan’s adults aged >60 years.

## 2. Methods

### 2.1. Study Design and Participants

This cross-sectional study was conducted in several polyclinics and geriatric service care centers in two cities (Almaty and Aktobe) of Kazakhstan from March to May 2024. A total of 670 respondents were interviewed. After excluding individuals with incomplete questionnaires (*n* = 28), the final number of participants was 642 (Figure 1). All respondents were interviewed by geriatricians and general practitioners using questionnaires that included the BI, the simple frailty questionnaire (FRAIL scale), the Mini Nutritional Assessment (MNA), The Strength, Ambulation, rising from a Chair, Stair Climbing, and History of Falling Questionnaire (SARC-F), the Mini-Cog, and the Fracture Risk Assessment Tool (FRAX). Body mass index (BMI) and grip strength were also measured. The participants were recruited from among all patients eligible for inclusion living in a medical or social institution (n = 278). In the city polyclinics of Almaty and Aktobe, patients who came to a general practitioner’s appointment from March to May 2024 were recruited (n = 364). These medical institutions were selected based on the place of work of the main researchers. All patients were familiarized with the study and signed an informed consent form.

Including a comparison between Almaty and Aktobe—two cities with distinct socioeconomic profiles—can enrich the contextual interpretation of the findings. Almaty, as Kazakhstan’s largest and most economically developed city, offers more extensive healthcare and social services infrastructure, while Aktobe, a growing regional center, presents a contrasting picture with more limited access to specialized elder care and support systems. These differences may influence patterns of functional decline, care-seeking behavior, and access to LTC resources among older adults [12].

The inclusion criteria were individuals aged 60 years and above who agreed to participate in the study. The exclusion criteria were older patients with severe and decompensated diseases that would have impeded the study; those with concomitant oncological or severe psychiatric conditions that would have impeded the study (schizophrenic spectrum disorders, severe depressive disorders with psychotic symptoms, delirium or acute confusion, moderate to severe intellectual disability); and older people who did not understand the purpose of the study. All the questionnaires and informative agreements were in two languages: Kazakh and Russian. The data on patients were for 642 individuals. The required sample size for this cross-sectional study was estimated using Epi Info, based on a reported prevalence of functional dependence of 27.8% among older adults in Asia [14]. Using a 95% confidence level and a margin of error of 5%, the minimum sample size was calculated to be 309 participants. To account for an anticipated 10% non-response rate or incomplete data, the target sample size was increased to 340 participants.

### 2.2. Data Collection and Study Variables

The data collected for this study included demographic data: age, sex, nationality, and place of living (home or nursing home).

Several independent variables were selected and used for the different phases of analysis performed for this study. These variables were chosen based on the previous literature showing their associations with functional decline in institutionalized persons [15].

Functional status was assessed by the BI. Proposed guidelines for interpreting Barthel scores are that scores of 0–20 indicate “total” dependency, 21–60 indicate “severe” dependency, 61–90 indicate “moderate” dependency, 91–99 indicate “slight” dependency, and 100 indicates “total” independence [16].

BMI is a measure of body fat based on height and weight that applies to adult men and women. For adults, BMI falls into these categories: below 18.5 (underweight); 18.5–24.9 (normal); 25.0–29.9 (overweight); 30.0 and above (obese). Based on previous research, we divided the selection into 4 categories [17].

Frailty was assessed according to the definition proposed by Fried [18]. Specifically, frailty was defined as having at least three of the following five criteria: exhaustion, low physical activity, slowness, unintentional weight loss, and low grip strength. Frailty state was assessed using the FRAIL scale, consisting of fatigue, resistance (defined as the ability to climb one flight of stairs), ambulation (walk one block), number of comorbid illnesses <5, and weight loss of more than 5% in the previous year. One point was given if the patients answered yes to each question, and the results were scored as follows: non-frail (score of 0), pre-frail (score of 1 or 2), and frail (score 3–5) [19]. Based on previous research, we divided the selection into 2 categories: non-frail and pre-frail in the category “no frail”, and frail in the category “frail” [20].

SARC-F is a self-completed questionnaire that includes five items based on the cardinal signs of sarcopenia. These five components are strength, assistance with walking, getting up from a chair, climbing stairs, and falling. The score varies from 0 to 10, and for each item 0 to 2 points are given. Sarcopenia is indicated by a score as follows: normal muscle function (score of 0–3), risk of sarcopenia (score of 4–10) [21].

The MNA involves rapid assessment of nutritional status in older patients in outpatient clinics, hospitals, and nursing homes. The MNA test is composed of simple measurements and brief questions that can be completed in about 10 min. Discriminant analysis was used to compare the findings of the MNA with the nutritional status determined by physicians, using the standard extensive nutritional assessment including complete anthropometric, clinical biochemistry, and dietary parameters. The sum of the MNA score distinguishes between older patients with normal nutritional status (score > 23.5), risk of malnutrition (score of 17–23.5), and malnutrition (score of 0–16) [22]. Based on previous research, we combined the selection into 2 categories: normal nutritional status and risk of malnutrition in the category “normal”, and malnutrition in the category “malnutrition” [23].

The Mini-Cog© was developed to help identify, in non-specialist settings, individuals likely to have clinically significant cognitive impairment. It was constructed by combining 3-word recall and a clock drawing task, included as an executive/cognitive composite. The scoring for cognitive screening was as follows: no cognitive impairment (score of 3–5) and risk of cognitive impairment (score of 0–2) [24].

FRAX shows a 10-year probability of hip fracture and major osteoporotic fractures in patients based on bone mineral density and clinical risk factors [25]. The FRAX model was created for the Republic of Kazakhstan, based on a regional population assessment of the incidence of hip fractures [26]. The FRAX scale for the Kazakhstan platform has the following scores (without densitometry): low risk (score of 0–19.9), high risk (score over 20.0).

Handheld Dynamometry (HD) is a method utilized to assess grip strength. The dynamometer is taken in hand with the dial inside. The arm is withdrawn from the body until a right angle is obtained with it. The second arm is released down along the body. The dynamometer is compressed with maximum force for 3–5 s. For more accurate results, it is recommended to measure the strength of a handshake three times for the right and left hands. The rest time between sets is at least 30 s. Grip strength was evaluated using the average or maximum value of the strongest handshake on the left and right. The score for low dynamometry was ≤29 kg for men and ≤17 kg for women [27].

Categories of functional status were divided in two categories: total dependency and severe dependency in category (1), “dependent patients”; and moderate dependency, slight dependency, and total independence in category (2), “active patients”. The prevalence of dependent patients was obtained by calculating the proportion of patients who were categorized as severe or totally dependent divided by the total study subjects. For the statistical analysis, frailty status was divided into (1) non-frail and (2) frail. Categories of age group were as follows: (1) 60–74 years old and (2) 75 and older. The subjects were categorized based on their sex as male or female. Categories of nationality were as follows: (1) Kazakh and (2) others. Categories of location status were as follows: (1) nursing home and (2) home. Categories of BMI were as follows: (1) underweight, (2) normal, and (3) overweight and obesity. Categories of cognitive function were as follows: (1) no cognitive impairment and (2) risk of cognitive impairment. Categories according to nutritional status were as follows: (1) normal and (2) malnutrition. Categories according to risk of fracture were as follows: (1) low and (2) high. Categories according to the presence of sarcopenia were as follows: (1) yes and (2) no.

We translated and validated all scales using generally accepted recommendations. All patients filled out the Kazakh and Russian versions of all scales. Using the Cronbach’s alpha coefficient, a high internal consistency was found, equal to 0.96.

### 2.3. Statistical Analysis

Data analysis was performed using jamovi (Version 2.6). The data are reported as mean ± standard deviation (SD) with 95% confidence intervals. Two group comparisons (dependent patients vs. active patients) were conducted using the independent samples Student’s *t*-test for all continuous outcome measures, and a chi-squared test for all categorical outcome measures. The Bonferroni correction was applied to adjust for multiple comparisons by dividing the standard *p*-value (0.05) by the number of variables (15) analyzed. The statistical significance level was 0.003. Variables that were significantly associated between groups were then combined into a multivariate binomial logistic regression analysis to assess the association between severe and total functional dependence and the independent variables. The association between risk factors and severe and total functional dependence was summarized with adjusted odds ratios (AORs) and their 95% confidence intervals (CIs), obtained from logistic regression.

## 3. Results

The primary data on patients were for 642 individuals. Of these, patients recruited from a Social Services Center in Almaty numbered 278, those from a polyclinic in Almaty numbered 148, and those from a polyclinic in Aktobe numbered 216.

We collected data from different medical care centers in Kazakhstan. The male-to-female ratio for the study subject was nearly 1:1. A total of 642 participants were included in this study (304 males and 338 females), age 75.28 ± 8.74 years, weight 71.88 ± 13.09 kg, height 1.65 ± 0.09 m, and BMI 26.27 ± 4.44 Kg/m^2^, of which 278 (43.3%) were identified as dependent patients. On average, dependent patients were older (71.2%) (*p* < 0.001) than active patients. There were more females (60.1%), than males (*p* < 0.001), more Kazakhs (61.2%) (*p* = 0.003) than patients of other nationalities, and a higher proportion of patients who lived at home (82.4%), than older adults in nursing homes (*p* < 0.001). A total of 86.3% of dependent patients were frail (*p* < 0.001), and 78.4% had a high risk of fracture and osteoporosis (*p* < 0.001). More than 59% of dependent patients had malnutrition (*p* < 0.001). On the other hand, a larger proportion of dependent patients did not have sarcopenia (57.2%) and did not have cognitive impairment (56.1%) (*p* < 0.001). Table 1 provides the full demographic, anthropometric, and clinical information for the populations studied.

Table 2 presents the multivariable logistic regression analysis results, highlighting significant factors contributing to severe and total functional dependence. The odds of having severe and total functional dependence are significantly higher for frail participants (AOR = 2.96, 95% CI [1.70, 5.16], *p* < 0.001) compared to non-frail participants. The odds of having severe and total functional dependence at home are eleven times higher (AOR = 11.90, 95% CI [5.77, 24.55], *p* < 0.001) than in nursing homes. The odds of having severe and total functional dependence are significantly two times higher for participants with a probability of sarcopenia (AOR = 2.61, 95% CI [1.49, 4.55], *p* < 0.001) compared to those with no sarcopenia. The odds of having severe and total functional dependence are significantly three times higher for participants with high risk of fracture (AOR = 3.30, 95% CI [1.94, 5.61], *p* < 0.001) compared to those with low risk. The odds of having severe and total functional dependence are significantly higher for participants with low dynamometry (AOR = 1.05, 95% CI [1.03, 1.07], *p* < 0.001) compared to those with normal dynamometry.

## 4. Discussion

Among the 642 older people in Kazakhstan in this study, 43.3% were dependent patients and 56.7% were active patients. Severe and total functional dependence is significantly associated with old age, low dynamometry, frailty, being at home, high risk of fracture and osteoporosis, and sarcopenia.

The BI is a widely used observer-based instrument to measure physical function. The structural validity, reliability, and interpretability of the BI are considered sufficient for measuring and interpreting changes in physical function of geriatric rehabilitation patients [28]. This is confirmed by many previous studies. Researchers from Canada determined that the ADL score was the strongest single predictor of functional decline trajectory that residents followed [29]. Similar studies have already been conducted in Japan [30]. Other researchers found differences in supposed boundary scores in ADLs, as well as in the prevalence of diseases in past medical histories between certified healthy elderly people and specified or certified elderly people [31]. In the next study, the BI was confirmed as a useful tool for evaluating functional abilities, quality of life, and depressive state in community-dwelling older adults aged 65 years and older [32]. Researchers from Japan carried out a cross-sectional study examining a correlation between BI scores and LTC. There was good correlation between the BI scores and LTC and care need levels overall and in several subgroups [33].

Our research reinforces the global consensus that sarcopenia, frailty, and grip strength are closely related to functional decline in aging populations. The results of this study largely align with previous research, demonstrating significant correlation between BI dependence and frailty. Previous work has shown this by identifying frailty using indicators such as old age and functional status. This study also found that frail patients are at a higher risk of disability and functional dependence [34]. A recent study analyzed the data of 2907 adults aged 70–84 years in Korea, where severe and moderate functional dependence were significantly associated with frailty [35]. Another study is a randomized controlled trial developed in non-geriatric hospital inpatient settings in Spain, Italy, and the United Kingdom. The main results of this study included correlation between the BI and frailty at 3 months [36].

Correlation between the BI and age is the same as in a previous study. In a cohort study, it was proven that there is an age-dependent relationship between the BI and sociodemographic characteristics relating to dependency [37].

Our finding of a strong correlation between the BI and low dynamometry is consistent with previous studies. A multicenter cross-sectional study was conducted on institutionalized older adults, consisting of the SARC-F questionnaire, handgrip strength assessment, and the BI [38]. Another study in South Australia showed the relationship between the BI and HD indicators [39]. A Brazilian study conducted across a wide range of participants revealed a clear and consistent correlation between grip strength and functional independence, which aligns with our own observations [40].

Our research links fracture and osteoporosis risk directly to functional dependence. Similarly, the association between the BI and osteosarcopenia is a plausible one. According to the cross-sectional analysis of geriatric inpatients, the BI was lower for sarcopenic and osteosarcopenic subjects [41]. Furthermore, research on the rehabilitation of hip fractures (n = 127, with an average age of 81) revealed that sarcopenia tripled the likelihood of incomplete recovery (odds ratio of 3.07) and was associated with lower BI scores upon discharge [42]. Our research in Kazakhstan significantly expands upon these findings by connecting the risk of fractures to the initial level of functional dependence, whereas previous studies primarily concentrated on the outcomes after a fracture.

In the physical domain, we found a correlation between sarcopenia and the BI. Likewise, a study conducted among Turkish elderly men living in the community (n = 274, average age approximately 74 years) found that a low calf circumference (< 31 cm), which is an indirect indicator of sarcopenia, was strongly associated with dependence in ADLs and IADLs (approximately 22–47%) [43]. A study conducted in Eastern Turkey involving 404 elderly individuals revealed a high prevalence of sarcopenia (46.8%) and frailty (37.4%). Sarcopenia was strongly associated with both dependence in ADLs and IADLs and an increased risk of falls (odds ratio of 7.4 for falls) [44]. Results of a pre–post mixed-methods study showed significant correlation between the BI and sarcopenia at baseline and after 6 months [45]. In the next study, patients with sarcopenia had significantly reduced life autonomy (determined by the BI; median 55 vs. 60 points, *p* < 0.001), meaning that sarcopenic patients were less autonomous in their daily life [46]. These results align closely with our findings on sarcopenia and frailty causing functional dependence.

According to the results of this study, the odds of having severe and total functional dependence at home were higher than in a nursing home. However, many studies have shown the opposite results [29,30]. Perhaps this was due to the fact that the nursing home where patients were recruited is a medical and social institution for people who need housing. The main criterion for staying in this institution is not health indicators, but a lack of housing. Another explanation for these results may be that the facility is well equipped and provides good care for residents (balanced nutrition, exercise, and medical supervision).

There were some limitations in our study. First, the number of participants was small, so statistical significance was not achieved for all indicators. Second, the influence of all confounding factors was not eliminated. We did not assess concomitant diseases, polypharmacy, or bad habits; thus, we cannot rule out a certain degree of residual confounding.

This study also had some strengths, which include its prospective design and the use of standardized procedures and validated instruments for the physical exam. To the best of our knowledge, this study is the first study to use the BI in Kazakhstan.

## 5. Conclusions

In conclusion, this research aimed to evaluate functional autonomy and identify the factors that contribute to it among the elderly population in Kazakhstan, with the goal of improving long-term care planning. Our findings indicate that advanced age, low grip strength, frailty, sarcopenia, increased risk of fractures and osteoporosis, and being at home are all significantly associated with severe or complete functional dependence. These findings provide clarity to our research objectives by identifying key biomedical and social factors that predict a decline in daily functioning. By integrating these factors into a multifaceted framework, we offer a comprehensive risk profile that is relevant to geriatric care in Kazakhstan. Unlike many previous studies, we emphasize the importance of homebound status and fracture risk alongside traditional indicators such as sarcopenia and frailty, providing context-specific insights that complement international evidence.

These results emphasize the critical importance of early detection, preventive measures, and personalized interventions to preserve functional autonomy in aging individuals. The research also underscores the significance of incorporating functional assessment into LTC policies and community health initiatives to reduce dependency and improve the quality of life of Kazakhstan’s aging population. This work defines risk factors of severe and total functional dependence in older people, and has important practical implications. Detection and prevention of any of the factors should be emphasized in clinical practice for better outcomes for older patients and lower healthcare costs. Future research should establish the specific mechanisms of the association between more risk factors and LTC.

## Figures and Tables

**Figure 1 healthcare-13-01878-f001:**
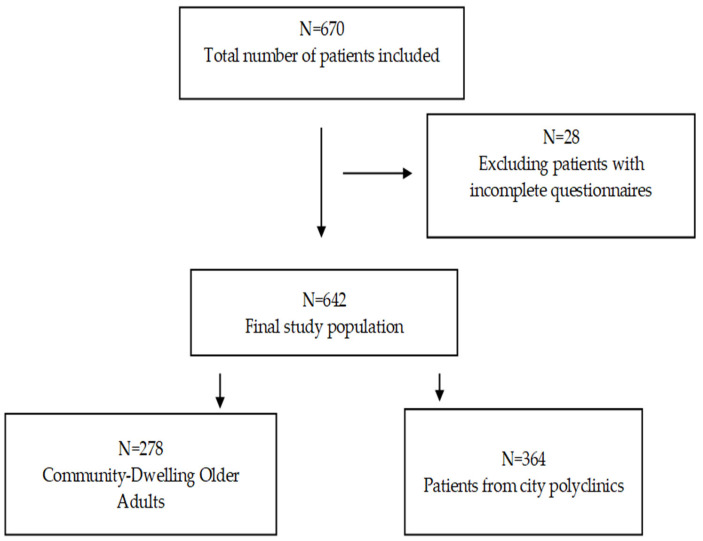
Participant flow.

**Table 1 healthcare-13-01878-t001:** Demographic, anthropometric, and clinical characteristics of patients (N = 642).

Characteristics	Total, *n* = 642 (100%)	Active Patients, *n* = 364 (56.7%)	Dependent Patients, *n* = 278 (43.3%)	*p*-Value
Age (years, mean ± standard deviation (SD))	75.28 ± 8.74 (60–100)	71.9 ± 7.90	79.7 ± 7.83	<0.001 *
Weight (kg, mean ± SD)	71.88 ± 13.09 (41–125)	73.0 ± 13.4	70.4 ± 12.5	0.011
Height (m, mean ± SD)	1.65 ± 0.09 (1.41–1.90)	1.65 ± 0.09	1.64 ± 0.08	0.087
Body mass index (BMI) (Kg/m^2^, mean ± SD)	26.27 ± 4.44 (16–48)	26.5 ± 4.80	26.0 ± 3.93	0.191
Dynamometry (kg, mean ± SD)	31.5 ± 14.9 (0–87)	36.9 ± 16.6	27.4 ± 11.9	<0.001 *
Sex (n, %)				<0.001 *
Male	304 (47.4%)	193 (53.0%)	111 (39.9%)	
Female	338 (52.6%)	171 (47.0%)	167 (60.1%)	
Classification by age (n, %)				<0.001 *
60–74 years	319 (49.7%)	239 (65.7%)	80 (28.8%)	
75 and older	323 (50.3%)	125 (34.3%)	198 (71.2%)	
Nationality (n, %)				0.003 *
Kazakhs	349 (54.4%)	179 (49.2%)	170 (61.2%)	
Others	293 (45.6%)	185 (50.8%)	108 (38.8%)	
Place of observation (n, %)				<0.001 *
Community-dwelling older adults	278 (43.3%)	229 (62.9%)	49 (17.6%)	
Home	364 (56.7%)	135 (37.1%)	229 (82.4%)	
BMI (n, %)				0.064
Underweight	12 (1.9%)	10 (2.8%)	2 (0.7%)	
Normal	251 (39.1%)	139 (38.2%)	112 (40.3%)	
Overweight	277 (43.1%)	149 (40.9%)	128 (46.1%)	
Obesity	102 (15.9%)	66 (18.1%)	36 (12.9%)	
Fracture Risk Assessment Tool (FRAX) (n, %)				<0.001 *
Low risk	300 (46.7%)	240 (65.9%)	60 (21.6%)	
High risk	342 (53.3%)	124 (34.1%)	218 (78.4%)	
Frailty questionnaire (FRAIL) (n, %)				<0.001 *
Not frail	238 (37.1%)	200 (54.9%)	38 (13.7%)	
Frail	404 (62.9%)	164 (45.1%)	240 (86.3%)	
Mini Nutritional Assessment (MNA) (n, %)				<0.001 *
Normal	360 (56.1%)	247 (67.9%)	113 (40.6%)	
Malnutrition	282 (43.9%)	117 (32.1%)	165 (59.4%)	
The Strength, Ambulation, rising from a Chair, Stair Climbing, and History of Falling Questionnaire (SARC-F) (n, %)				<0.001 *
No sarcopenia	433 (67.4%)	274 (75.3%)	159 (57.2%)	
Risk of sarcopenia	209 (32.6%)	90 (24.7%)	119 (42.8%)	
Mini-Cog (n, %)				<0.001 *
No cognitive impairment	408 (63.6%)	252 (69.2%)	156 (56.1%)	
Risk of cognitive impairment	234 (36.4%)	112 (30.8%)	122 (43.9%)	

* indicates a statistically significant difference at *p* < 0.003. Continuous variables were compared using the independent samples *t*-test. Categorical variables were analyzed using the chi-squared test. A *p*-value less than 0.003 was considered statistically significant after Bonferroni correction. BMI—body mass index; FRAIL—simple frailty questionnaire; SARC-F—The Strength, Ambulation, rising from a Chair, Stair Climbing, and History of Falling Questionnaire; MNA—Mini Nutritional Assessment; FRAX—Fracture Risk Assessment Tool; SD—standard deviation.

**Table 2 healthcare-13-01878-t002:** Multivariable logistic regression analysis of factors associated with severe and total functional dependence.

Variables	AOR	95% CI	*p*-Value
Age	1.1348	(1.0746, 1.198)	<0.001 *
Age classification 75 and older	0.7181	(0.3102, 1.662)	0.439
Sex Women	1.3498	(0.7966, 2.287)	0.265
Nationality Kazakhs	1.1781	(0.6628, 2.094)	0.576
Place of observation Home	11.904	(5.774, 24.55)	<0.001 *
Dynamometry Low	1.0522	(1.0304, 1.074)	<0.001 *
FRAIL Have frailty	2.9589	(1.6971, 5.159)	<0.001 *
Mini-Cog Cognitive impairment	2.1796	(1.2358, 3.844)	0.007
MNA Malnutrition	1.1846	(0.7077, 1.983)	0.519
SARC-F Probability of sarcopenia	2.6053	(1.4906, 4.554)	<0.001 *
FRAX High risk	3.3016	(1.9447, 5.605)	<0.001 *

The reference category is “active patients”. * indicates a statistically significant difference at *p* < 0.003. Multivariable logistic regression analysis was used to identify independent factors associated with severe and total functional dependence. AORs with 95% CI and *p*-values are reported. A *p*-value < 0.003 was considered statistically significant. FRAIL—simple frailty questionnaire; SARC-F—The Strength, Ambulation, rising from a Chair, Stair Climbing, and History of Falling Questionnaire; MNA—Mini Nutritional Assessment; FRAX—Fracture Risk Assessment Tool; AOR—adjusted odds ratio; CI—confidence interval.

## Data Availability

The raw data supporting the conclusions of this article will be made available by the authors, without undue reservation.

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
