# Peer review of "Assessing Functional Independence and Associated Factors in Older Populations of Kazakhstan: Implications for Long-Term Care"

_healthcare, 2025, doi:10.3390/healthcare13151878_

Round 1
Reviewer 1 Report
Comments and Suggestions for Authors
I think this is an interesting paper.
With some improvements to the following points, readers will better understand the researchers’ intended meaning.
I hope this helps.
Major revision
- The title and aim of the research are not aligned.
- Clearly state the research question of this study.
Is it that "BI can predict the risk of frailty or sarcopenia in patients requiring long-term care (LTC)?" This could not be understood from the analysis results.
- I think it would also be useful to illustrate (make figure) what the researchers are trying to clarify in this research at the methods section.
- In the Discussion section, the authors should discuss the results and how they can be interpreted from the perspective of previous studies and the working hypotheses or research questions. Currently, it seems that they are merely citing previous research.
5.Please make a conclusion section. Explain how your research aims and questions have been clarified.
Minor revision
- Acronyms/Abbreviations/Initialisms should be defined the first time they appear in each of the three sections: the abstract; the main text; the first figure or table. When defined for the first time, the acronym/abbreviation/initialism should be added in parentheses after the written-out form.
- This journal Healthcare is not APA style. Please read the Instructions for Authors. In the text, reference numbers should be placed in square brackets [ ], and placed before the punctuation; for example [1], [1–3] or [1,3]. For embedded citations in the text with pagination, use both parentheses and brackets to indicate the reference number and page numbers; for example [5] (p. 10). or [6] (pp. 101–105).
- References should be described as follows, depending on the type of work: Journal Articles: 1. Author 1, A.B.; Author 2, C.D. Title of the article. Abbreviated Journal Name Year, Volume, page range.
- Line 152: “LTC patients” this expression should be changed to “patients who are receiving LTC.”
- Line 196: “Muscle strength” should be revised to “grip strength.”
- Lines 224-237: The information provided here should be described in the research methods section.
- Line 240: Kg/m2 should be revised to kg/m². Also, in Table 1.
- In Table 1, dose dynamometry means grip strength?
- In Table 1, term elderly and senile. Examples of bias-free language for age. Problematic and preferred examples are presented along with explanatory comments. Terms for older adults, Problematic: the elderly, elders, elderly people, the aged, aging dependents, seniors, senior citizens. Preferred: older adults, older people, persons aged 65 years and older. Please revise.
- Tables 1 and 2. Please define any abbreviations used in the table, below each table.
- The AORs and 95% confidence intervals in Table 2 should be corrected to two decimal places.
- In Table 2. Is “95% AOR” a 95% confidence interval (CI)?
- Lines 201 to 203, this sentence can edit like below: “Grip strength was evaluated using the average or maximum value of the strongest handshake on the left and right.”
Author Response
Major revision
Comments 1: The title and aim of the research are not aligned.
Response 1: Thank you for pointing this out. We agree with this comment, and offer the following explanation. We changed title on "Assessing Functional Independence and Associated Factors in Elderly Populations of Kazakhstan: Implications for Long-Term Care" . The aim of the research: find the current prevalence of patients with severe and total functional dependence and its associated factors among people 60 years and over in Almaty and Aktobe cities of Kazakhstan.
Comments 2: Clearly state the research question of this study. Is it that "BI can predict the risk of frailty or sarcopenia in patients requiring long-term care (LTC)?" This could not be understood from the analysis results.
Response 2: Research question of this study: "What is the prevalence of of patients with severe and total functional dependence among adults aged 60 years and over in Almaty and Aktobe, Kazakhstan, and what demographic, clinical, and functional factors (frailty, sarcopenia, grip strength, fracture risk, osteoporosis, and living arrangement) are significantly associated with this need?"
Comments 3: I think it would also be useful to illustrate (make figure) what the researchers are trying to clarify in this research at the methods section.
Response 3: Thank you for pointing this out. We agree with this comment, and offer the following explanation. The discussion of these point is added “Figure 1. Participant flow” to the methods section.
Comments 4: In the Discussion section, the authors should discuss the results and how they can be interpreted from the perspective of previous studies and the working hypotheses or research questions. Currently, it seems that they are merely citing previous research.
Response 4: Thank you for pointing this out. We agree with this comment, and offer the following explanation. The discussion of these point is added this information to the discussion section in lines 337-353, 366-385 and 389-398.
Comments 5: Please make a conclusion section. Explain how your research aims and questions have been clarified.
Response 5: Thank you for pointing this out. We agree with this comment, and offer the following explanation. The discussion of these point is added this information to the conclusion section in lines 407-427.
Minor revision
Comments 1: Acronyms/Abbreviations/Initialisms should be defined the first time they appear in each of the three sections: the abstract; the main text; the first figure or table. When defined for the first time, the acronym/abbreviation/initialism should be added in parentheses after the written-out form.
Response 1: Thank you for pointing this out. We agree with this comment, and offer the following explanation. We corrected it.
Comments 2: This journal Healthcare is not APA style. Please read the Instructions for Authors. In the text, reference numbers should be placed in square brackets [ ], and placed before the punctuation; for example [1], [1–3] or [1,3]. For embedded citations in the text with pagination, use both parentheses and brackets to indicate the reference number and page numbers; for example [5] (p. 10). or [6] (pp. 101–105).
Response 2: Thank you for pointing this out. We agree with this comment, and offer the following explanation. We corrected it.
Comments 3: References should be described as follows, depending on the type of work: Journal Articles: 1. Author 1, A.B.; Author 2, C.D. Title of the article. Abbreviated Journal Name Year, Volume, page range.
Response 3: Thank you for pointing this out. We agree with this comment, and offer the following explanation. We corrected it.
Comments 4: Line 152: “LTC patients” this expression should be changed to “patients who are receiving LTC.”
Response 4: The authors acknowledge the reviewer’s comment and agree with the reviewer. We realized that in this study it is possible to determine only the current prevalence of patients with severe and total functional dependence and related factors among people aged 60 years and older in Kazakhstan. We have replaced the designation “long-term care needs” in the manuscript with “severe and total functional dependence” (lines 44, 51 and 251-252).
Comments 1: Line 196: “Muscle strength” should be revised to “grip strength.”
Response 1: The authors acknowledge the reviewer’s comment and agree with the reviewer. We revised this word in lines 243.
Comments 5: Lines 224-237: The information provided here should be described in the research methods section.
Response 5: The authors acknowledge the reviewer’s comment and agree with the reviewer. The discussion of these point is added this information to the methods section in lines 251-264.
Comments 6: Line 240: Kg/m2 should be revised to kg/m². Also, in Table 1.
Response 6: The authors acknowledge the reviewer’s comment and agree with the reviewer. We revised it in line 288 and in Table1.
Comments 7: In Table 1, dose dynamometry means grip strength?
Response 7: Thank you for pointing this out. Yes. Handheld Dynamometry is a method utilized to assess grip strength in kg.
Comments 8: In Table 1, term elderly and senile. Examples of bias-free language for age. Problematic and preferred examples are presented along with explanatory comments. Terms for older adults, Problematic: the elderly, elders, elderly people, the aged, aging dependents, seniors, senior citizens. Preferred: older adults, older people, persons aged 65 years and older. Please revise.
Response 8: The authors acknowledge the reviewer’s comment and agree with the reviewer. The discussion of these point is removed terms “elderly and senile” in Table1.
Comments 9: Tables 1 and 2. Please define any abbreviations used in the table, below each table.
Response 9: The authors acknowledge the reviewer’s comment and agree with the reviewer. We defined abbreviations below each table.
Comments 10: The AORs and 95% confidence intervals in Table 2 should be corrected to two decimal places.
Response 10: Thank you for pointing this out. Yes.
Comments 11: In Table 2. Is “95% AOR” a 95% confidence interval (CI)?
Response 11: Thank you for pointing this out. Yes. We changed “95% AOR” on “95% CI” in Table2.
Comments 12: Lines 201 to 203, this sentence can edit like below: “Grip strength was evaluated using the average or maximum value of the strongest handshake on the left and right.”
Response 12: The authors acknowledge the reviewer’s comment and agree with the reviewer. We changed this sentence in lines 248-250.

Reviewer 2 Report
Comments and Suggestions for Authors
General Comments
The overall use of English is not up to standard. Extensive English editing is necessary throughout the manuscript.
Abstract
- The sentence “43.3% were patients, who need long-term care; and 56.7% were active patients” should be rephrased for clarity. It is unclear what distinguishes “patients” from “active patients.” This distinction must be explained clearly in the abstract. Abstract should be standalone without requiring readers to refer to the main text.
- The abbreviation "AOR" should be defined at first mention.
- The term “low dynamometry” is vague and should be clarified.
Introduction
4. Line 99: Please compare the elderly population in Kazakhstan with the global trends.
5. Are there any similar studies conducted in other populations or countries? A brief discussion of their findings and the rationale for conducting this study in the Kazakhstani context would help justify the study.
6. The research gap and the significance of this study are not clear. Please elaborate on why this study is needed and what contribution it makes.
Methods
7.How were participants recruited? Please describe the recruitment method in more detail.
8.A major concern is the exclusion of “people with neurological and mental conditions that impede the study.” This is problematic because individuals with neurological conditions often represent a large proportion of those requiring long-term care. This exclusion may significantly affect the generalizability of the findings. Please explain why the study team decided to exclude this group of patients.
9. Defining individuals with a Barthel Index (BI) score of 0–60 as needing LTC may not be appropriate unless this is supported by national policy or guidelines. The BI only assesses ADL function and does not fully capture long-term care needs. Long-term care needs are multidimensional and influenced by factors such as comorbidities, environmental risks, social support, and coping ability.
10.Additional background on the two study cities, Almaty and Aktobe, particularly regarding their socioeconomic profiles would enhance contextual understanding.
11.Line 138: The sample size estimation is inadequately described. Please provide more details
Results
12. Line 214: “significantly associated.” Are you referring to statistical difference between groups?
13. It is surprising to find that staying in nursing homes are less likely to receive long-term care. Conceptually, nursing homes are considered a form of long-term care.
14. Line 246: This sentence appears incorrect reporting of the results: “A larger proportion of LTC patients was not sarcopenia (57.2%) and had low likelihood dementia (56.1%).”
Discussion & conclusion
15 Once again, I remain unconvinced that LTC needs can be determined solely based on BI score categories.
Author Response
Reviewer 2
Abstract
Comments 1: The sentence “43.3% were patients, who need long-term care; and 56.7% were active patients” should be rephrased for clarity. It is unclear what distinguishes “patients” from “active patients.” This distinction must be explained clearly in the abstract. Abstract should be standalone without requiring readers to refer to the main text.
Response 1: The authors acknowledge the reviewer’s comment and agree with the reviewer. The discussion of these point is added to the abstract section in lines 47-50.
Comments 2: The abbreviation "AOR" should be defined at first mention.
Response 2: The authors acknowledge the reviewer’s comment and agree with the reviewer. We added definition to the abstract section in line 52.
Comments 3: The term “low dynamometry” is vague and should be clarified.ent countries?
Response 3: The authors acknowledge the reviewer’s comment and agree with the reviewer. We added this information to the abstract section in line 60.
Introduction
Comments 4: Line 99: Please compare the elderly population in Kazakhstan with the global trends.
Response 4: The authors acknowledge the reviewer’s comment and agree with the reviewer. The discussion of those points is added to the discussion section in lines 123-131.
Comments 5: Are there any similar studies conducted in other populations or countries? A brief discussion of their findings and the rationale for conducting this study in the Kazakhstani context would help justify the study.
Response 5: The authors acknowledge the reviewer’s comment and agree with the reviewer. The discussion of these point is added to the introduction section in lines 99-111.
Comments 6: The research gap and the significance of this study are not clear. Please elaborate on why this study is needed and what contribution it makes.
Response 6: The authors acknowledge the reviewer’s comment and agree with the reviewer. The discussion of these point is added to the introduction section in lines 99-111.
Methods
Comments 7:How were participants recruited? Please describe the recruitment method in more detail.
Response 7: The authors acknowledge the reviewer’s comment and agree with the reviewer. The discussion of these point is added to the methodology section in lines 158-171.
Comments 8: A major concern is the exclusion of “people with neurological and mental conditions that impede the study.” This is problematic because individuals with neurological conditions often represent a large proportion of those requiring long-term care. This exclusion may significantly affect the generalizability of the findings. Please explain why the study team decided to exclude this group of patients.
Response 8: The authors acknowledge the reviewer’s comment and agree with the reviewer. We conveyed our message incorrectly. Patients were excluded from the study if they were diagnosed with psychiatric conditions that could significantly affect cognitive information processing, comprehension, or the ability to reliably participate in assessment. For example: schizophrenic spectrum disorders, severe depressive disorders with psychotic symptoms, delirium or acute confusion, moderate to severe mental retardation. These conditions were excluded to ensure the validity of self-assessment or indicators based on tools such as the Barthel index, FRAIL scale, Mini-Cog and MNA, which require a minimum level of cognitive function and cooperation. We have replaced these words on “severe psychiatric conditions” in lines 181-184.
Comments 9: Defining individuals with a Barthel Index (BI) score of 0–60 as needing LTC may not be appropriate unless this is supported by national policy or guidelines. The BI only assesses ADL function and does not fully capture long-term care needs. Long-term care needs are multidimensional and influenced by factors such as comorbidities, environmental risks, social support, and coping ability.
Response 9: The authors acknowledge the reviewer’s comment and agree with the reviewer. We realized that in this study it is possible to determine only the current prevalence of patients with severe and total functional dependence and related factors among people aged 60 years and older in Kazakhstan. We have replaced the designation “long-term care needs” in the manuscript with “severe and total functional dependence”.
Comments 10: Additional background on the two study cities, Almaty and Aktobe, particularly regarding their socioeconomic profiles would enhance contextual understanding.
Response 10: The authors acknowledge the reviewer’s comment and agree with the reviewer. The discussion of these point is added to the methodology section in lines 172-178.
Comments 11: Line 138: The sample size estimation is inadequately described. Please provide more details
Response 11: The authors acknowledge the reviewer’s comment and agree with the reviewer. The discussion of these point is added to the methodology section in lines 188-191.
Results
Comments 12: Line 214: “significantly associated.” Are you referring to statistical difference between groups?
Response 12: Yes. We added this information in line 276.
Comments 13: It is surprising to find that staying in nursing homes are less likely to receive long-term care. Conceptually, nursing homes are considered a form of long-term care.
Response 13: Thank you for pointing this out. We agree with this comment, and offer the following explanation. The discussion of these point is added to the discussion section in lines 391-398.
Comments 14: Line 246: This sentence appears incorrect reporting of the results: “A larger proportion of LTC patients was not sarcopenia (57.2%) and had low likelihood dementia (56.1%).”
Response 14: Thank you for pointing this out. We agree with this comment, and offer the following explanation. We corrected this sentence in line 295 and Table 1.
Discussion & conclusion
Comments 15: Once again, I remain unconvinced that LTC needs can be determined solely based on BI score categories.
Response 15: Thank you for pointing this out. We agree with this comment, and offer the following explanation in Response 9.

Round 2
Reviewer 1 Report
Comments and Suggestions for Authors
Thank you very much for the corrections.
Minor revision
…Kazakhstan[1].
…of LTC by 2030[2].
Please insert a space between Kazakhstan and [1].
…Kazakhstan <insert space> [1].
…of LTC by 2030<insert space>[2].
Please make similar corrections in the revised manuscript.
Background/Objectives section of the abstract can be improved as follows.
Accurately assessing the independence level of older adults using useful assessment tools is an important step toward providing them with the necessary care while preserving their dignity. These tools allow older adults to receive effective, personalized home care, which improves their quality of life. This study aimed to clarify the current prevalence of severe and complete functional dependence and the associated factors among Kazakhstan’s older adults aged >60 years.
Introduction section, lines 150-154, can be improved as follows.
Accurate dependency assessments using useful assessment tools are essential for providing effective, personalized home care that improves the quality of life for older adults. In this context, developing, implementing, and evaluating innovative long-term care (LTC) models in geriatric practice are paramount.
This study aimed to determine the prevalence of severe and complete functional dependence and its associated factors among Kazakhstan’s adults aged >60 years.
Please indicate the statistical methods used in this study under the Tables 1 and 2. This will make it easier for readers to understand the analyzed tables.
I wish you continued success in your research and in providing high-quality medical care.
Again, thank you so much.
Author Response
Comments 1: …Kazakhstan[1].
…of LTC by 2030[2].
Please insert a space between Kazakhstan and [1].
…Kazakhstan <insert space> [1].
…of LTC by 2030<insert space>[2].
Please make similar corrections in the revised manuscript.
Response 1: The authors acknowledge the reviewer’s comment and agree with the reviewer. We have corrected the manuscript according to the remark.
Comments 2:Background/Objectives section of the abstract can be improved as follows.
Accurately assessing the independence level of older adults using useful assessment tools is an important step toward providing them with the necessary care while preserving their dignity. These tools allow older adults to receive effective, personalized home care, which improves their quality of life. This study aimed to clarify the current prevalence of severe and complete functional dependence and the associated factors among Kazakhstan’s older adults aged >60 years.
Response 2: The authors acknowledge the reviewer’s comment and agree with the reviewer. We improved it in line 43-48.
Comments 3: Introduction section, lines 150-154, can be improved as follows.
Accurate dependency assessments using useful assessment tools are essential for providing effective, personalized home care that improves the quality of life for older adults. In this context, developing, implementing, and evaluating innovative long-term care (LTC) models in geriatric practice are paramount.
This study aimed to determine the prevalence of severe and complete functional dependence and its associated factors among Kazakhstan’s adults aged >60 years.
Response 3: The authors acknowledge the reviewer’s comment and agree with the reviewer. We improved it in line 153-158.
Comments 4: Please indicate the statistical methods used in this study under the Tables 1 and 2. This will make it easier for readers to understand the analyzed tables.
Response 4: The authors acknowledge the reviewer’s comment and agree with the reviewer. The discussion of those points is added under tables in lines 301-302 and 325-327.
